# What Makes a Rabbit Cute? Preference for Rabbit Faces Differs according to Skull Morphology and Demographic Factors

**DOI:** 10.3390/ani9100728

**Published:** 2019-09-26

**Authors:** Naomi D. Harvey, James A. Oxley, Giuliana Miguel-Pacheco, Emma M. Gosling, Mark Farnworth

**Affiliations:** 1School of Veterinary Medicine and Science, The University of Nottingham, Sutton Bonington Campus Leicestershire LE12 5RD, UK; giuliana_miguel@hotmail.com; 2Independent Researcher, Measham, Swadlincote Derbyshire DE12 7LQ, UK; james_oxley1@hotmail.com; 3APHA Scientific, Sutton Bonington, Loughborough, Leicestershire LE12 5RB, UK; emma.gurney@hotmail.co.uk; 4School of Animal Rural & Environmental Sciences, Nottingham Trent University, Brackenhurst Campus, Southwell, Nottinghamshire NG25 0QF, UK; mark.farnworth@ntu.ac.uk

**Keywords:** rabbit, pet, brachycephaly, preference, domestication, cephalic index

## Abstract

**Simple Summary:**

The majority of pet rabbits have flatter, shorter faces than wild rabbits. However, rabbits with flat faces are at higher risk of developing considerable health problems, including painful dental problems. The aim of this research was to identify which type of rabbit face people actually prefer, in order to better understand why rabbits with flat faces might be bred and purchased. Images of 25 rabbit faces were assigned to a face-type group by 134 veterinary professionals. Through an online questionnaire, people then rated each of the 25 images according to preference for the rabbits’ faces, and a total of 20,858 questionnaires were completed globally. Through statistical modelling, we show for the first time that mildly flat-faced rabbits (on a scale of extremely flat-faced to extremely long-faced) are more preferred globally than any other kind, and that the longest faced rabbits are the least preferred. Aside from the shape, other features of rabbit faces that were preferred include a soft, medium-light fur appearance and being generally short-furred. These results support the theory that the human preference for the baby-like features of flat-faced rabbits has driven their popularity. We would encourage breeders to avoid breeding extremely flat-faced rabbits due to the associated health problems and to focus instead on breeding more preferred mildly flat-faced, erect-eared rabbits such as the Havana breed.

**Abstract:**

Domesticated rabbits typically exhibit shorter, flatter skulls than their wild counterparts (brachycephalism). However, brachycephaly is associated with considerable health problems, including problems with dentition. The aim of this study was to establish which type of rabbit face people prefer, with a particular emphasis on skull morphology and brachycephaly. We grouped 25 images of rabbit faces by cephalic degree based on ratings assigned by 134 veterinary professionals. An online questionnaire was then launched, in which people could rate each of the 25 images according to preference for the rabbits’ faces, and a total of 20,858 questionnaires were completed globally. Repeated-measure, multi-level general linear modelling revealed mildly-brachycephalic rabbits to be the most preferred type of rabbit, and moderately-dolichocephalic (longer skulled) rabbits to be the least preferred. The preference for brachycephalic rabbits was stable across continents, and as such it is highly plausible that human preference has been a driver for the shortening of the skull typically seen in domestic rabbits, perhaps as a result of the ‘baby-schema’. Additional features of rabbit faces that were preferred include a soft, medium-light fur appearance and being generally short-furred. These novel insights may prove useful in the improvement of the public understanding of rabbit health and welfare. The relationship between preference and skull shape is particularly pertinent to future work concerning rabbit health, given the cross-species evidence that having a flat face is associated with chronic health conditions.

## 1. Introduction

Recent research estimates that over a quarter of a million pet rabbits are bred annually in the United Kingdom (UK) for online sales alone [1]. Within this number, the two most common breeds (Miniature-lop and Netherland dwarf) comprise 53.1% of the population, and both breeds are moderately to extremely brachycephalic (BC) [1], a trait characterised by a foreshortening of the skull. Indeed, domesticated rabbits typically exhibit shorter, flatter skulls than their wild counterparts [2], meaning that brachycephalism has become the new norm for domesticated breeds; but what has driven this change has yet to be established. Certain morphological features such as white spots, lop-ears and skull morphology changes (both elongated and shortened) have been shown to occur spontaneously in various species when they are selected for behavioural ‘tameness’ during the domestication process (see [3] for an overview of this phenomenon). However, additional mechanisms, such as secondary selection for morphological traits, are likely to have been involved for any of these features to become predominant amongst domestic populations, as can be seen here with domestic rabbits and brachycephaly. When considering the health and welfare implications of companion animal breeding, most research and public discourse concerns the impacts on cats and dogs. For example, only a single study has been published about the breeding of pet rabbits [1], despite them being the third most popular companion animal (excluding fish) in the UK [4].

Where breed standards are used to classify and select dogs for specific morphological traits, they are known to contribute directly [5] or indirectly [6] to numerous health-related disorders. Whilst there are breed standards for rabbits (e.g., [7]), there is considerably more variation within a breed in the pet population (as opposed to show rabbits) than is commonly seen for dogs or cats (authors’ personal observations). Despite this intra-breed variation, certain rabbit breed-types are artificially selected for phenotypic traits similar to those seen in other companion animal species. Homologous features across species are manipulated through selective breeding, in some cases leading to extreme variations. For example, both the Miniature-lop rabbit and the Bassett hound dog are identified by large drooping ears (amongst other traits). Understandably, changes in the homologous features of companion animals have common health implications. For example, being lop-eared is thought to increase the likelihood of otitis in rabbits [8,9], just as breeds of dog with pendulous and hairy ears are overrepresented in cases of pseudomonas otitis [10]. Additionally, lop-eared rabbits are at a higher risk of dental problems compared to erect-eared rabbits due to associated changes in the skull and jaw shape [9,11].

Perhaps one of the most extreme morphological changes within companion animal breeds is brachycephaly, which results in wholesale changes to the jaw and skull, in turn affecting the orbits and cranial cavity. The majority of domestic rabbits have shorter skulls than their wild counterparts, some more extreme than others, which has a noted impact on mastication performance, predisposing them to dental health problems [2]. Brachycephaly in companion animals is associated with considerable health problems, including problems with dentition (evident in rabbits [2,12,13], cats [14], and dogs [15]), the respiratory system (evident in cats [16] and dogs [17,18]) and the eyes (evident in cats [19] and dogs [20]). Whilst a general lack of studies on rabbit health in relation to conformation means that published evidence is minimal, veterinarians report that BC rabbits also exhibit problems with the respiratory system similar to BC dogs [21], particularly when under anaesthesia [22], and encounter problems associated with distorted tear ducts [23]. Despite this common problem with brachycephaly, BC traits/breeds are growing in popularity amongst companion animal owners, as evidenced by both UK and Australian pedigree dog registrations [24,25].

One of the possible drivers for the popularity of, and selection for, BC animals is the baby-schema (Kindchenschema) [26]. First proposed by ethologist Konrad Lorenz, the baby-schema outlines the concept that baby-like features, including larger eyes, bulging craniums and shorter noses, make an individual appear more vulnerable and that these features release a motivation in people to care for them. Indeed, studies have shown that there is a strong relationship between ‘cuteness’ and the baby-schema, which directly elicits positive affective reward circuits in people [27,28]. The baby-schema has also been shown to extend to the human-animal relationship, with children responding to the baby-schema in both humans and animals (cats and dogs) from an early stage in development [29].

Although the factors driving rabbit breed selection are relatively unexplored, dog breed popularity is partly driven by social influences and fashion [30,31]. BC dog and cat breeds are now well represented within the top ten most commonly owned breeds (e.g., 4/10 of cat breeds in the United States of America [32] and 3/10 of dog breeds in the UK [33]). Despite an absence of the same systematic data collection, this trend is also apparent within the companion rabbit population, with moderately to extremely BC rabbit breeds and lop-eared breeds comprising just over half the population [1], and it will likely have the same societal drivers. Research suggests a positive relationship between dog breed popularity and the number of inherited disorders within that breed [31]. To compound this problem further, motivational factors reported by dog and cat owners indicate that appearance is a stronger impetus, and health a lesser one, for those choosing a BC breed as compared to a non-BC breed [34,35,36].

For cats, although images of BC individuals have been shown to be less preferred when compared to their mesocephalic (a head of medium proportions, not markedly brachycephalic or dolichocephalic) and dolichocephalic (having a relatively long skull, with a length longer than it is wide) counterparts, substantial preferences for BC cats persisted in those individuals that currently owned a BC breed [37]. Again, these preferences seem to run contrary to data that demonstrate that BC cats, especially those with an extreme shortening of the muzzle, experience comparatively poorer health outcomes [16]. Once owned, BC cat owners do not consider their cats to be as healthy as other cat owners and are less likely to recommend their BC breed to others [35], yet this does not seem to impact owner preference for BC cat faces. However, there is no data yet that investigates people’s preference for rabbit faces.

Given the relative paucity of data concerning domestic rabbits, this study sought to explore if and how preferences for rabbit facial structures may mirror those already established in cats and dogs. If the baby-schema is the main driver for preferences for rabbit faces, we would expect to see higher preference ratings for BC rabbits compared to mesocephalic (MC) or dolichocephalic (DC) rabbits. Alternatively, if rabbits are considered the same as cats when it comes to facial feature preference, we would expect to see higher scores for mesocephalic rabbits, and significantly lower scores for rabbits with extreme facial conformations. Considering the popularity of BC rabbit breeds in the pet population and the considerable health risks of extreme brachycephaly, particular emphasis will be placed on investigating the factors that influence preference for extremely-BC rabbits.

## 2. Materials and Methods

Photographic images of rabbit faces were collected by the authors, including the authors’ own rabbits and images that were freely available on the internet under creative commons licences (for a full list of photograph sources, see Appendix A). All images were originally in colour and were changed to black and white to avoid any potential influence of the background colour. Each image was trimmed to focus on the head of the animal with as little of the background included as possible. Care was taken to try and select a set of images that covered the full range of skull shapes present within the pet rabbit population, including representations of rabbits with differing phenotypic features (long and short hair, lop and non-lop eared) within each type of skull shape from BC, to MC and DC (see Appendix B, Figure A1 for all 25 images).

Two online surveys were designed using the platform SurveyGizmo (www.surveygizmo.com) based upon methods presented in Farnworth et al. [37], who used questionnaires to investigate how phenotypic attributes, including the skull shape, affected the preferences of cat owners. The first survey used here was developed to ascertain the veterinary judgements of the skull morphology for each of the 25 rabbit images, in order to score them on a cephalic scale, as there is no validated cephalic index for rabbits. The second survey was developed to collect data relevant to preference ratings for each image from the wider public. Each survey is described in turn below.

The study, including both surveys, was approved by the University of Nottingham Research Ethics Committee (reference no. 2284180501). Both surveys were anonymous, collecting no personal data. Participants gave their consent by ticking a box associated with a statement regarding the anonymous nature of the survey, which meant that we would not be able to delete anyone’s data upon request. At the end of each survey, participants were, however, given the option to confidentially share an email address with us for the purpose of informing them of the results upon publication. Consent was gained here (again by ticking an unchecked box) specifically for the purpose of storage at the University of Nottingham (accessible only to N.D.H) and the use of this email only to disseminate the results, after which the emails would be deleted.

### 2.1. Veterinary Survey and Cephalic Index

Due to the varying nature of the photographic images (i.e., photographs taken from different distances, angles and varying image qualities) and inclusion of long-haired rabbit breeds (e.g., Lionhead), physical measurements were unable to be consistently taken from the images. Therefore, in order to create a cephalic index, an anonymous veterinary survey was designed for veterinary professionals to rate the 25 rabbit facial images (see Appendix C, Table A1 for a full list of the questions asked). The survey included four sections: (i) introductory page, aim of the study, length of survey, informed consent and contact details; (ii) veterinary information (i.e., are they currently a practicing veterinarian/vet nurse/vet student, country of residence and work, are they a rabbit specialist and frequency of rabbit examinations); (iii) cephalic ratings for the 25 rabbit facial images from a scale that ranged from 1 to 7 (1 = extreme brachycephalism (BC), 2 = moderate BC, 3 = mild BC, 4 = mesocephalic (MC), 5 = mild dolichocephalism (DC), 6 = moderate DC and 7 = extreme DC) (based on Farnworth et al. [37]). The terms were defined as follows on the first page of the cephalic ratings section:Brachycephalic: having a head with a markedly flattened, short face;Mesocephalic: having a head of medium proportions, not markedly brachycephalic or dolichocephalic;Dolichocephalic: having a relatively long skull, with a length longer than it is wide.

The questionnaire was reviewed by all authors prior to distribution. No validation was conducted. A link to the final questionnaire was distributed directly to three UK veterinary schools (Royal Veterinary College; Bristol Veterinary School, University of Bristol; School of Veterinary Science and Medicine, University of Nottingham). It was also disseminated through social media (Facebook and Twitter) and remained live for a period of four weeks from the 21st of May to the 18th of June, 2018.

### 2.2. Public Preference Ratings

An anonymous questionnaire was developed based on a previous study by Farnworth et al. [37] that was open for anyone to complete, whether they owned a rabbit or not, no matter where in the world they resided or what their age was. The questionnaire consisted of four sections, including (i) introductory page (explained the aim and length of the survey, informed consent and further contacts in case a respondent had any questions), (ii) demographic information (i.e., age group, gender, country of residence of respondent, highest education level and if they work in a veterinary or animal care profession), (iii) pet rabbit ownership (i.e., current and past rabbit ownership, current and previously owned rabbit breeds and length of ownership), and (iv) preference ratings for each of the 25 rabbit face images. Each image was scored on an 11-point Likert scale that ranged from 0 to 10, with 0 being “I do not like this rabbit at all” and 10 being “This is my favourite type of rabbit”. If a respondent scored an image at the low end of the scale (between 0 and 3) or the high end of the scale (7–10), this was taken to mean that they disliked/liked the rabbit in the image (respectively), and a further multiple-choice question was asked regarding what it was they disliked/liked about the rabbit (see Appendix C, Table A2 for a full list of questions asked and answer options). For both the public and veterinary survey, the order in which the images appeared was the same and started with a wild rabbit, followed by an image presentation in a pseudorandom order (BC-MC-DC).

The questionnaire was written in English and was also translated into Spanish (translation by G.M.P). The survey was distributed from the 6th of July to the 28th of August 2018, through social media (Facebook and Twitter) and a University of Nottingham press release.

### 2.3. Statistical Analysis

All data distributions between groups were visually checked to ensure that the distributions were comparable (for example that none were bimodal) as per best practice recommendations [38] using the R package *beanplot* [39]. Overall, due to the large sample size, significance was considered at *p* < 0.001, a stricter threshold than is standard.

The median cephalic rating provided to each rabbit image by the veterinary consensus was used in the analysis to group the images according to their cephalic type. Images were also assigned to phenotypic categories dependent on whether the rabbits in them were lop-eared or not, had all short fur or areas of long fur, and whether they appeared to be coloured solid white, solid dark, solid grey, solid medium-light, or patterned.

Univariable associations between phenotypic features (cephalic rating treated as groups, coat colour, fur type, and ear type) and preference scores were analysed using Kruskal-Wallis (for variables with three or more categories) or Mann-Whitney tests (for variables with two categories). To investigate how respondent-related variables (age, rabbit owner/non-rabbit owner, experience working in an animal-related profession, continent, and highest education level) may be associated with preferences for extremely-BC rabbits, Mann-Whitney or Kruskal-Wallis tests were used as appropriate for testing for associations with scores given to the rabbits classified as extremely-BC only (those with a cephalic median of 1).

Phenotypic features and respondent variables that were significant to < 0.2 were taken forwards [40] into multi-level models with the respondent ID and image ID included as nested repeated measures. Data were analysed in MLwiN (version 3.03 [41]): one model was used to compare the phenotypic features against preference ratings for all animals, and a second model was used to compare the respondent factors with the preference for rabbits categorised as extremely-BC.

## 3. Results

### 3.1. Veterinary Cephalic Consensus

Over a four-week period, the veterinary survey received 134 completed responses from veterinary professionals, of which 16% (21) considered themselves to be rabbit specialists and another 16% were not sure if they classified as such (Table 1). Sixty-nine of the veterinarians (90%) were currently practicing in clinics, and when asked how often they treated rabbits in clinics, 29 (42%) said they saw rabbits at least once a day, 24 (35%) at least once a week, 13 (19%) at least once a month and three (4%) less than or once every six months.

Six of the images were assigned a median cephalic rating of 1—extreme brachycephalism, three were assigned a median of 2—moderate brachycephalism, six achieved a median rating of 3—mild brachycephalism, five were assigned a median of 4—mesocephalic, three images were assigned a median of 5—mild dolichocephalism, two were assigned a median of 6—moderate dolichocephalism, and no images achieved a median of 7—extreme dolichocephalism (Table 2). Interestingly, Images 24 and 25, which were of wild rabbits, were assigned a median cephalic rating of 5 (mild dolichocephalism), implying that the general view of a mesocephalic rabbit (where mesocephalic was defined as being of medium proportions) was one that has a muzzle shorter than that of a wild rabbit. Furthermore, despite trying to capture the full range of extreme brachycephalism to extreme dolicocephalism, 15 (60%) of the 25 images were classified as BC to at least some extent.

### 3.2. Preference Study Participants

A total of 20,858 complete responses to the preference questionnaire were received internationally (Table 3). Country of residence was answered by 15,613 respondents (75%), the majority of which (54%) were from North America or Europe (37%). Female participants were over-represented in the sample, comprising 67% of all respondents who provided details on their gender, whilst 16% were male, and 17% were gender non-conforming or had transitioned genders. The majority of participants (68%) were aged between 18 and 34, had never worked in a profession related to animal care/science (93%), were educated to an undergraduate level or higher (54%) and had never owned a rabbit (70%).

### 3.3. Associations between Preference Ratings and Phenotypic Features

All phenotypic variables were significantly associated with preference ratings (see Appendix A for all univariate results). The preference ratings differed significantly between the cephalic groups in the univariate analysis (Kruskal Wallis statistic = 26,841, df = 5, *p* < 0.001), with the highest scores given to rabbits in the Mildly-BC group and the lowest given to those in the Moderate-DC group (cephalic ratings 3 and 6, respectively, in Figure 1 and Appendix A). The individual rabbit that received the highest preference ratings overall was number 13, which is a Havana rabbit rated as Mildly-BC (see Appendix A).

When focussing on rabbits rated as the participants’ “favourite type of rabbit” (images that were assigned a top score of 9 or 10), the group of rabbits least likely to receive a ‘top’ preference rating were those rated Moderately-DC. The most likely to be rated 9 or 10 were images of Mildly-BC rabbits, which were 3.31 times more likely to receive a top preference rating compared to the images in the Moderately-DC group (Table 4). Meanwhile, those in the Moderately-BC and Extremely-BC groups were 2.84 and 1.97 times more likely, respectively, to receive a top preference rating.

Once all the phenotypic variables were controlled within a repeated-measures multi-level analysis, all phenotypic variables (cephalic group, ear type, fur type, and colour type) contributed to the preference ratings (Table 5) without altering the previously described associations with the cephalic group. Rabbits with non-lop ears and short fur were preferred more than rabbits with lop ears and long fur. Rabbits that appeared grey or with mixed colours were rated as less preferred than dark rabbits, whilst rabbits with light and medium light colours were rated as more preferred than rabbits with a dark colour.

### 3.4. Factors Associated with Preference Ratings for Extremely Brachycephalic Rabbits

A univariate analysis revealed significant associations for every demographic variable (see the Appendix A for the full univariate results). Table 6 presents the results of the multivariate analysis comparing the demographic population characteristics against the preference ratings for Extremely-BC rabbit images, controlling for image as a repeated measure. People who reported living in a country located either in Africa, Asia, Latin America and the Caribbean, or in North America gave Extremely-BC rabbits higher preference ratings than people living in countries in Europe or Oceania. When looking at the age of the respondent, the preference ratings got lower as the age increased. Respondents who were not currently working in, or had never worked in, a career related to animal health gave higher preference scores for Extremely-BC rabbits than those currently working in a career related to animal health. An increasing education level was typically associated with reductions in preference ratings for Extremely-BC rabbits. Finally, current rabbit owners gave Extremely-BC rabbits higher preference scores than non-rabbit owners did.

### 3.5. Reasons for High and Low Ratings of the Most and Least Preferred Cephalic Groups

When considering the most preferred type of rabbits, those in the Mildly-BC group, softness (45.7%), face shape (42.5%), and fluffiness (36.7%) were the main reasons given for assigning a high preference score (Figure 2 and Figure 3). Being too fluffy (32.0%) and the face shape (24.1%) were the top-cited reasons for disliking Mildly-BC rabbits. For the least preferred type of rabbit, the Moderately-DC rabbits, the face shape, eyes and erect ears were the main reasons given for both disliking and liking these rabbits, with ‘looking soft’ only being a strong factor for preference (38.9%). For Extremely-BC rabbits, the most common reasons given for giving a high score was the rabbit looking soft (46.3%), its fluffiness (41.5%), its face shape (40.9%) and its eyes (39.4%). In contrast, the main reasons people gave for giving a low score to Extremely-BC rabbit images were the rabbit’s eyes (25.5%), it being too fluffy (22.4%), and it having erect ears (19.9%), whilst face shape was only selected as a reason for providing a low score by 5.4% of the people, and only 8.1% of people said they disliked lop-ears in these rabbits. A past association with a similar looking rabbit was given as a reason for liking a rabbit by just 4.7% to 7.9% of question respondents and was given as a reason for disliking a rabbit by 0.3% to 1.0% of question respondents.

The face shape was 7.5 times less likely to be cited as a factor in why people disliked rather than liked Extremely-BC rabbits (Figure 3). However, this effect reversed as brachycephalism became less marked, with people more likely to dislike a rabbit’s face as the skull became more elongated.

The ear form (lop or erect) did not appear to have a clear association with the skull length. Preference for ear type was found to increase in some skull forms, (e.g., erect ears were more likely to be cited as a factor that is important for preference than for disliking in the Mildly-BC and Moderately-DC images) but had converse outcomes for other images (e.g., in the Extremely-BC images, lop-ears were more likely to be cited as a factor that is important for preference than for disliking).

Colour type was far more likely to be selected as a reason for liking than disliking a rabbit across all cephalic groups, but particularly for the Mildly-BC group. Eyes were 4 times more likely to be selected as a reason for liking a Mildly-BC rabbit than disliking one of these rabbits and looking soft was the most impactful reason overall for liking any group of rabbit, with face shape coming in second. Rabbits were not likely to be disliked for not being fluffy, but could be disliked for being too fluffy, and a past experience with similar looking rabbits had very little (conscious) impact on whether people liked or disliked an image.

## 4. Discussion

Domesticated rabbits typically exhibit shorter, flatter skulls than their wild counterparts, a difference suggested to be due, in part, to differences in diet [2]. Here, we demonstrate for the first time that when asked to judge a rabbit based purely on its facial appearance, globally, people exhibit a distinct preference for rabbits with flatter, brachycephalic skull shapes and that extremely-BC face shapes are less likely to be disliked than longer rabbit faces. Such preferences may explain the typical brachycephaly observed in contemporary domesticated rabbits.

This study received an exceptional level of engagement with the public, with 20,858 people spread over every continent completing the preference questionnaire and rating all 25 rabbit images. Through this exceedingly rich dataset, we were able to show that mildly-BC rabbits are the most commonly preferred rabbits, closely followed by moderately-BC rabbits, whilst the least preferred types of rabbits according to skull morphology are moderately-DC rabbits, followed by extremely-BC rabbits. Although extremely-BC rabbits were the second lowest scoring group, their face shape was the least often ‘disliked’ factor of their appearance.

When investigating the demographic factors that influence preference for rabbits with extremely-BC faces, we showed that the preference scores are relatively stable across continents, with only participants from Europe and Oceania differing from the rest of the world by scoring these images significantly lower overall in the multivariable models (Table 4). In comparison, a similar study investigating owners’ preference for cat faces reported greater differences between continents for extremely-BC cat faces, although participants from Europe and Oceania assigned extremely-BC cats the lowest ratings [37]. Although the same rating scale was used here, these differences between studies may be due to the difference in species evoking different emotional responses. The rabbits rated in this study were assigned higher preference scores overall compared to the cats rated in Farnworth et al., with the lowest median preference score achieved here being 5 by moderately-DC rabbits, compared to extremely-BC cats in [37] receiving a median preference score of 3. Additionally, mesocephalic cats were found to be the most preferred facial type in [37], compared to rabbits, where mildly-BC forms were the most preferred. The tendency for people from Europe to rate the extremely-BC forms of both species lower than people from elsewhere in the world could be due to cultural differences in perceptions, which may be linked to media coverage and campaigning for awareness of issues associated with brachycephaly (see, for example, Vets Against Brachycephalism [42]).

With the face morphology seen to be an important factor in the scoring of all rabbit types, it is likely that this, along with other prevalent factors that include fluffiness, ear position and softness, will influence decisions when acquiring pet rabbits. Rabbits, commonly, are bought more impulsively than other domestic pets [4,43,44]; a study by Edgar and Mullan (2011) demonstrated that out of 52 rabbit owners, 18% decided to buy the rabbit on the same day, 10% decided to buy their rabbit within a week and only 29% took more than one month to make their purchasing decision [43]. Whilst we do not know what drives an individual to choose a specific rabbit when they purchase or adopt one, a recent review on the factors involved in acquiring pet dogs concluded that physical appearance is one of the most widely reported reasons for choosing a particular dog, in addition to its behaviour and health, as well as social influences such as media trends [45]. Furthermore, owners of BC cats were more likely to report appearance as a factor in their decision to purchase their cat than owners of non-BC cats were [35]. Here, by using static images, we can only address the influence of the physical appearance of a rabbit on peoples’ preference for it. Such preferences can be used to infer likely trends in breeding selection and purchasing behaviour and cannot account for the influence of other factors such as size, health and individual animal behaviour. However, these data collected here are highly relevant to the online sales of rabbits. Influencing factors such as behaviour and interactions are eliminated in online sales, and appearance becomes one of the main deciding factors, which is an important consideration with a quarter of a million rabbits estimated to be sold online per year in the UK alone [1].

Traditionally, rabbits are bought as pets for children [44]; therefore, adult and child perception of rabbit appearance is an important factor in driving rabbit purchasing decisions. Unique to this study was the effect of age, with older people typically assigning lower preference ratings to the Extremely-BC rabbit images than younger people did, although children were not surveyed here. This association with age could be related to cultural shifts such as the more recent development of ‘cute-culture’. Cute-culture or *kawaii* originated in Japan and is typically characterised by the use of images and cartoons of small fluffy creatures and infants [46]. However, if anyone compares toys and cartoons from past decades to contemporary ones, a trend for characters with larger eyes and features that are more infantile can be observed (N.D.H personal observation). In fact, the term ‘cute’ is closely tied to the concept of the baby-schema [27,28]. It is possible that a greater use of the baby-schema in popular culture could have influenced preference ratings for the rabbit images in this study. Additionally, research suggests that women are more receptive to baby-schema than men [47]. With appearance as the only factor in perceived cuteness, the baby-schema is likely to have influenced the preference ratings in this study, and the over-representation of women respondents may have introduced a bias toward higher ratings for BC rabbits.

One finding from this study was that people who work in an animal health-related profession or had an advanced degree were likely to provide lower median preference scores for extremely-BC rabbit images. This finding mirrors the research into preferences for BC cats by Farnworth et al. [37]. These associations are potentially due to a greater awareness of, or exposure to, the health problems commonly associated with extreme brachycephaly. This theory is supported by the fact that as people’s education levels increased, their preference for extremely flat-faced rabbits decreased. Despite being considered an ‘exotic’ species (in the UK at least) with specific health and welfare requirements that differ from more familiar species such as dogs and cats [48], rabbits are freely available to buy without any provision of information on how to care for their specific or breed-related health needs. Given the various reasons that are likely to influence purchasing behaviour, it may be possible that knowledge of the higher risk of health and welfare problems in BC rabbits (as discussed earlier), perhaps illustrated with stories directly from owners of extremely-BC rabbits, could help to mitigate a preference for their appearance at the point of sale. However, with appearance proving to be the driving factor, and health a minor factor, in people’s decisions to acquire BC dogs [34,36] and BC cats [35], a legislative approach toward stemming the breeding of animals with extreme conformations may be required, as has recently been introduced by the Dutch Government regarding the breeding of BC dogs [49]. Such decisions are based on a large research base for the negative welfare impact of brachycephalism on dogs. Although BC rabbits are considered to be more at risk of health and welfare problems, this area has not been fully researched for rabbits, and further veterinary evidence is required to identify which breeds or types of rabbits are most at risk of compromised welfare and the extent to which they are affected. Such evidence would help to create guidelines for breeding rabbits and for owners of rabbits that are at risk of conformation-related health problems.

When focusing on how people scored extremely-BC rabbits (those most at risk for conformation-related health issues), the face shape was commonly cited as a reason for having a strong preference for them (57%) but was not commonly cited as a reason for disliking them (7%). This finding differed from other types of rabbits, with face shape more likely to be disliked as skulls became elongated. Such a finding suggests that a preference for rabbits is in large part driven by their facial conformation, and that extreme brachycephalism is not widely considered to be a disliked trait, even by those who did not like those rabbits’ appearance at all. In fact, the main reason given for disliking the extremely-BC rabbits was being lop-eared (36%), a trait that is independent of the skull length, which suggests that the popularity of extremely-BC rabbits could increase if un-coupled from them being lop-eared. This finding may explain the popularity of the Netherland dwarf (an erect-eared extremely-BC rabbit breed) as the second most common breed in the UK [1].

As a web-survey largely distributed through social media, the data described here were collected through self-selection, which introduces its own form of bias [50]. In addition to this, distribution through participant sharing on social media is likely to introduce considerable ‘in-group’ bias that means our respondents may be more similar to each other than a random selection of internet users would be. However, the multi-national coverage, large sample sizes and repeated-measure multi-level modelling conducted in the analysis go some way toward addressing these concerns. It also must be acknowledged that all internet surveys such as this one are automatically biased toward internet users, who may not have the same views as non-internet users. Other limitations include the way in which we classified the education status and rabbit colour here. Due to international differences in education schemes, we cannot be sure people were always in equivalent categories, and due to the pictures being made greyscale we could only evaluate colourings/markings instead of colour itself. Furthermore, although every effort was made to ensure the consistency of facial images, there were still differences relating to the image size, quality and the image backdrop. Despite a pre-selection of the rabbit images to balance the phenotypic traits (ear type, colour and fur length) across the skull morphology spectrum, we were unable to locate suitable images for moderately-DC rabbits with dark coloured or long fur, and this category was represented by just two individuals, which may have impacted the preference scores for this group. The features examined here were also confounded by individual rabbits, as for example we did not manipulate images to change just one variable at a time by for instance giving the same rabbit lop or erect ears. Instead, each feature was exampled by a different individual, and we cannot capture how the combination of features impacted people’s preference. This was controlled for, to an extent, by having the rabbit ID as a random effect within the models; however, future studies may want to consider the impact of individual phenotypic features by focussing on one trait such as the skull length or ear type by manipulating images, instead of using images of different rabbits, which would control for the confounding effects of the individual and background.

Future research in this area would do well to develop a rabbit cephalic index. Cephalic indices have proven to be useful measures for investigating conformation-related problems in dogs (e.g., [51]), and they are calculated simply as the ratio of skull width to skull length. For rabbits, this calculation may miss crucial details of the skull morphology, such as the height and relative position of the eyes. Here, we used the assessment of veterinary professionals to classify the rabbit images according to a cephalic rating scale; however, the cephalic degree is a continuum, and categories such as brachycephalic, mesocephalic and dolichocephalic are arbitrary [52]. There was a much higher degree of variance in the cephalic ratings assigned amongst the veterinary professionals than we had expected to see, which could not be explained by type (veterinarian, student or veterinary nurse) (see Appendix A for descriptive statistics broken down by type). For example, the typical range in the cephalic ratings for the images assigned a median value of 1 (extremely-BC), was 1 to 3 (with 3 being mildly-BC), and two images in this group had a range that extended to 4, which meant that some veterinary professionals classified them as being mesocephalic. Although this problem was mitigated here by using median values over a large sample of professionals, this finding in itself highlights an additional issue in relation to how people consider rabbits in terms of their skull morphology; it is possible that the predominance of BC rabbits in the pet population has normalised their appearance for some individuals, so that they no longer recognise their skull shape as being ‘abnormal’. This could have impacts on how veterinary research is conducted if clinical records are to be used to gather data on health in relation to conformation, as a BC rabbit may not be recorded as such. An objective measurement, such as a rabbit cephalic index, would provide a gold-standard quantification for future studies related to rabbit skull morphology.

## 5. Conclusions

When considering ‘what makes a rabbit cute?’, the baby-schema effect would suggest that rabbits with more extreme-BC features would be rated as being more preferred. Here, we provide new evidence that mildly-BC rabbits are the most preferred type of rabbit considering skull morphology. Additionally, despite the popularity of lop-eared rabbits, we show that erect-eared rabbits tend to be more preferred. Given that BC and lop-eared features are also significantly associated with greater health risks that compromise animal welfare, we would strongly advocate breeding away from extreme-BC conformations in rabbits, and breeding fewer lop-eared breeds. Whilst the overall trend showed a preference of mild brachycephalism in rabbit faces, a large proportion of people still assigned strong preference scores to images of rabbits with extremely-BC facial conformations, with current rabbit owners more likely to prefer extremely-BC rabbits. The preference scores for BC rabbits was stable across continents, and as such it is highly plausible that this preference has been a strong driver for the shortening of the skull typically seen in domestic rabbit species compared to their wild counterparts. Additional features of rabbit faces that were preferred include a soft, medium-light fur appearance, as well as being generally short-furred. Knowing that people have a general preference for BC rabbits, targeted education campaigns could be developed to raise awareness of conformation-associated health issues and to suggest more moderate or mesocephalic alternatives for prospective rabbit owners.

## Figures and Tables

**Figure 1 animals-09-00728-f001:**
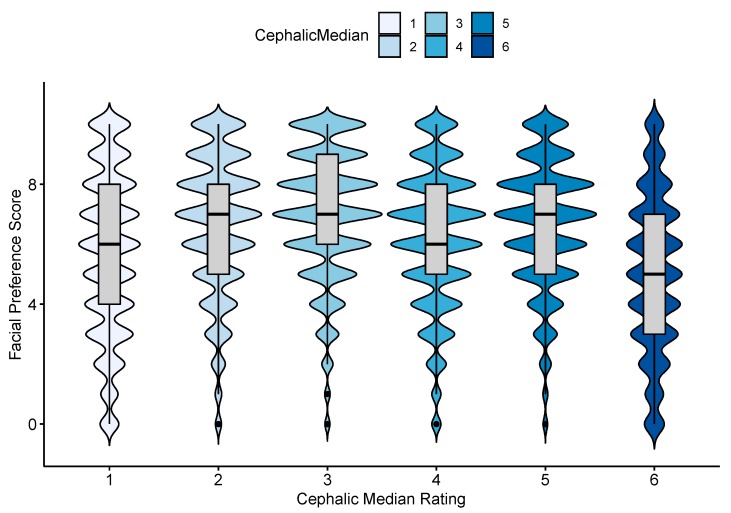
The distribution of the preference ratings assigned to rabbit face images by 20,858 people, within each of the six cephalic rating groups. The data are displayed as violin plots (where the width indicates the frequency of scores given) with boxplots inside showing the interquartile range and median. Cephalic groups range from 1 “Extremely-brachycephalic”, to 4 “Mesocephalic” and 6 “Moderate-dolichocephalic”. The preference ratings range from 0 (“I don’t like this rabbit at all”) to 10 (“This is my favourite type of rabbit”).

**Figure 2 animals-09-00728-f002:**
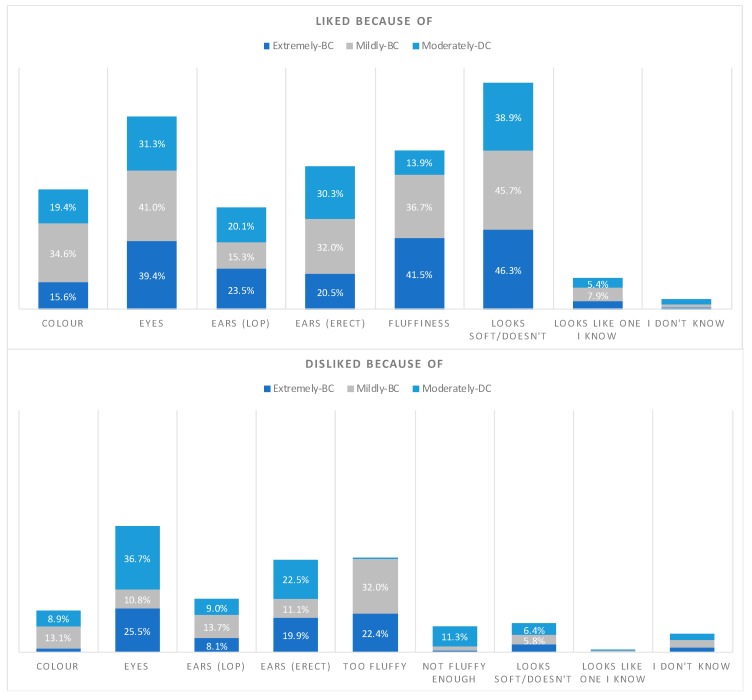
The proportion of times each reason (excluding Face Shape) was selected for assigning a high preference score (‘liking’, range 7–10) or low preference score (‘disliking’, range 0–3) to rabbits in the cephalic groups Extremely-BC, Mildly-BC and Moderately-DC (the *y*-axis has been made the same between both graphs to allow for comparison). Note that each person was able to select multiple reasons, so they were not mutually exclusive. The category ‘ears’ was calculated separately, depending on whether the rabbit in the image had lop or erect ears. For visual clarity, percentage values below 5.0% are not written.

**Figure 3 animals-09-00728-f003:**
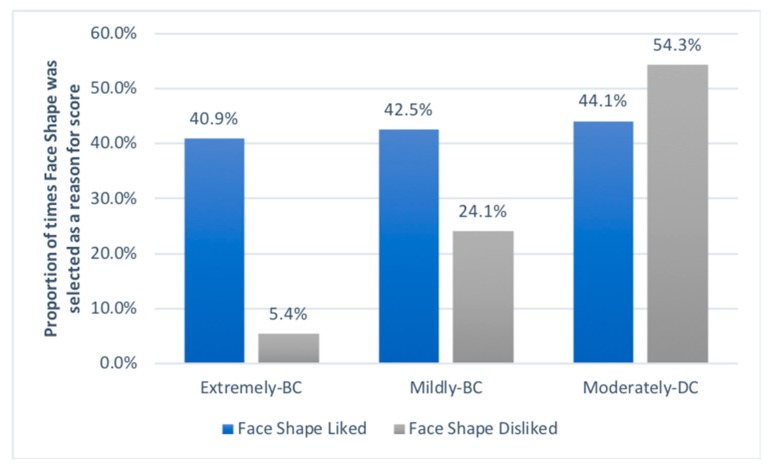
The proportion of times Face Shape was selected as a reason for assigning a high preference score (‘liking’, range 7–10) or low preference score (‘disliking’, range 0–3) to rabbits in the cephalic groups Extremely-BC, Mildly-BC and Moderately-DC.

**Table 1 animals-09-00728-t001:** Breakdown of the respondent type for the veterinary professional survey.

Type	N (% of Total)	Rabbit Specialist		
		Yes N (%)	Not Sure N (%)	No N (%)
Veterinarian	77 (57%)	13 (17%)	18 (23%)	46 (60%)
Veterinary Nurse	13 (10%)	3 (23%)	3 (23%)	7 (54%)
Veterinary Student	44 (33%)	5 (11%)	2 (4%)	37 (84%)
Total	134 (100%)	21 (16%)	23 (17%)	90 (67%)

**Table 2 animals-09-00728-t002:** Cephalic rating summary statistics provided by 134 veterinary professionals for 25 rabbit face images. The rating scale ranged from 1 to 7: 1 = extreme brachycephalism, 2 = moderate brachycephalism, 3 = mild brachycephalism, 4 = mesocephalic, 5 = mild dolichocephalism, 6 = moderate dolichocephalism, and 7 = extreme dolichocephalism. See Figure A1 for images associated with each ID. SD, standard deviation.

Image ID	Median	Mean	SD	Minimum to Maximum
1	1	1.43	0.60	1 to 3
2	1	1.29	0.50	1 to 3
3	2	1.70	0.73	1 to 4
4	1	1.49	0.80	1 to 5
5	1	1.43	0.58	1 to 3
6	1	1.51	0.61	1 to 3
7	2	2.16	0.84	1 to 5
8	1	1.47	0.74	1 to 4
9	3	2.72	0.87	1 to 6
10	4	3.93	1.06	2 to 7
11	2	2.22	0.80	1 to 5
12	3	2.82	0.85	1 to 5
13	3	3.12	0.80	1 to 4
14	3	3.28	0.84	1 to 5
15	3	2.69	0.86	1 to 4
16	4	3.81	0.96	2 to 6
17	4	4.15	1.11	2 to 7
18	6	6.02	1.02	4 to 7
19	4	3.93	0.97	2 to 4
20	5	4.78	1.02	3 to 7
21	6	6.09	1.04	2 to 7
22	3	3.24	1.02	1 to 6
23	4	4.78	1.02	3 to 7
24	5	5.01	1.02	3 to 7
25	5	4.78	1.06	3 to 7

**Table 3 animals-09-00728-t003:** The demographics of respondents (*N*) to the preference survey filtered by continent. Total *N* = 20,858. The valid numbers for the overall sample per question are contained in the 1st row (variable descriptor). The differences between the total responses for continent and the sum of other subcategories represent missing data (e.g., Missing answers = continent total – total for answers in category [e.g., Gender category = (cis)Male + (cis)Female + Other]).

Continent	Gender (*N* = 15,242)			Age (*N* = 20,821)			Animal Care Professional (*N*=19,990)		Education (*N* = 15,777)		Rabbit Owner (*N* = 20,858)		
	(Cis)Male (*N* = 2458)	(Cis)Female (*N* = 10,191)	Other (*N* = 2593)	17 or Less (*N* = 2229)	18–34 (*N* = 14,112)	35+ (*N* = 4480)	Yes/I was (*N* = 1429)	Never (*N* = 18,561)	Non-Undergrad (*N* = 7236)	Undergrad or Higher (*N* = 8541)	Current (*N* = 1960)	Past (not Now) (*N* = 4392)	Never (*N* = 14,506)
Europe (*N* = 5773)	1100	3974	397	369	3739	1661	295	5041	2215	3546	939	1842	2992
Africa (*N* = 62)	15	44	1	3	43	16	8	54	16	46	9	13	40
Asia (*N* = 238)	46	164	21	28	180	29	27	211	89	149	23	68	147
Oceania (*N* = 660)	81	463	99	67	420	173	92	568	313	347	88	208	364
North America (*N* = 8480)	1075	5114	2015	962	6074	1436	723	7337	4347	4119	821	2092	5567
Latin American and Caribbean (*N* = 400)	103	271	22	37	227	136	105	295	171	229	58	123	219
Not Given (*N* = 5245)	38	161	38	763	3429	1029	179	5055	85	105	22	46	5177

**Table 4 animals-09-00728-t004:** Logistic regression output comparing how frequent a top preference rating (9 or 10 on the scale) was assigned to a rabbit image within each cephalic group, as compared to the images in Group 6 (moderately dolichocephalic). Significance was considered at *p* < 0.001. OR, odds ratio. CI, confidence interval.

Cephalic Group	Wald	*p*-Value	OR	95% CI
1 (Extremely-BC)	1445.7	<0.001	1.97	1.92–2.06
2 (Moderately-BC)	3038.8	<0.001	2.84	2.74–2.95
3 (Mildly-BC)	4586.6	<0.001	3.31	3.20–3.42
4 (Mesocephalic)	1195.8	<0.001	1.89	1.82–1.96
5 (Mildly-DC)	1692.4	<0.001	2.13	2.13–2.30

**Table 5 animals-09-00728-t005:** The multilevel regression model results for associations between the preference ratings (*N* = 20,858) for images of 25 different rabbit faces and the phenotypic characteristics of the rabbit faces. *All images were presented in greyscale, so ‘colour’ indicates whether the rabbits’ fur appeared to be all uniformly light, medium-light, grey, dark or whether it had a mixed colouring. Significance was considered at *p* < 0.001.

Variable	Type	Coefficient	SE	Z	*p*-Value
Intercept		5.96	0.01	595.50	<0.001
Cephalic group	1 Extreme-BC	Reference			
	2 Moderate-BC	0.94	0.02	62.80	<0.001
	3 Mild-BC	1.52	0.01	151.80	<0.001
	4 Mesocephalic	0.67	0.01	55.83	<0.001
	5 Mild-DC	1.38	0.02	92.20	<0.001
	6 Moderate-DC	−1.15	0.02	−76.80	<0.001
Ear Type	Non-lop	Reference			
	Lop	−0.05	0.01	−6.25	<0.001
Fur Type	Short Haired	Reference			
	Long Haired	−0.37	0.01	−40.89	<0.001
Colour Type	Dark	Reference			
	Grey	−0.68	0.02	−45.60	<0.001
	Light	0.08	0.01	6.92	<0.001
	Medium-light	0.50	0.01	35.36	<0.001
	Mixed	−0.69	0.01	−57.33	<0.001

**Table 6 animals-09-00728-t006:** The multilevel regression model results for associations between the preference ratings (*N* = 20,858) for images of 6 rabbit faces classified as being extremely-brachycephalic and the demographic population characteristics. Significance was considered at *p* < 0.001. SE, standard error.

Variable	Type	Coefficient	SE	Z	*p*-Value
Intercept		5.11	0.05	101.80	<0.001
**Continent**					
	Europe	Reference			
	Africa	0.56	0.14	4.05	<0.001
	Asia	0.61	0.07	8.54	<0.001
	Latin America and the Caribbean	0.45	0.06	8.02	<0.001
	North America	0.38	0.02	19.95	<0.001
	Oceania	0.05	0.05	1.07	0.296
	Not Given	0.26	0.08	3.32	0.001
**Age**					
	17 or younger	Reference			
	18 to 24	−0.13	0.05	−2.80	0.005
	25 to 34	−0.27	0.05	−5.70	<0.001
	35 to 44	−0.55	0.05	−10.62	<0.001
	45 to 54	−0.95	0.06	−16.29	<0.001
	55 to 64	−1.28	0.07	−18.33	<0.001
	65 to 74	−1.33	0.11	−11.75	<0.001
	75 or older	−1.69	0.23	−7.36	<0.001
	Not Given	−0.52	0.30	−1.79	0.076
**Vet or Animal Care/Science Worker**					
	Yes	Reference			
	No, but I have in the past	0.63	0.05	13.30	<0.001
	No, never	0.73	0.04	20.31	<0.001
	Not given	0.43	0.33	1.30	0.197
**Current Rabbit Owner**					
	No	Reference			
	Yes	0.72	0.03	27.85	<0.001
**Education**					
	No qualifications	Reference			
	GCSE/Graduated high school	−0.10	0.05	−2.06	0.043
	College courses, no degree	−0.19	0.05	−3.84	<0.001
	Trade/Technical school	−0.14	0.07	−2.03	0.046
	Bachelor’s degree	−0.27	0.05	−5.32	<0.001
	Associate degree	−0.10	0.06	−1.55	0.128
	Advanced degree (Master’s, Ph.D., M.D.)	−0.39	0.05	−7.48	<0.001
	Not Given	0.36	0.24	1.55	0.124

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
