# Peer review of "What Makes a Rabbit Cute? Preference for Rabbit Faces Differs according to Skull Morphology and Demographic Factors"

_animals, 2019, doi:10.3390/ani9100728_

Round 1
Reviewer 1 Report
This is a useful paper that explains preferences for specific types of domestic rabbit. I am impressed with the number of survey respondents that the authors have managed to collect, and the thorough approach to data analysis.
The article would benefit from a thorough read-over to remove long and wordy sentences, and improve clarity of explanation in some areas. Better use of citations is needed in several areas to ensure that all arguments are fully supported.
Detail review below:
Simple summary
The simple summary needs to be reworded because it contains numerous descriptions of the rabbits that are not really quantified or explained. what’s the difference, for example, between a flat-faced rabbit and a midly-flat-faced rabbit? How has this been measured?
Line 17 and 18 appears to say the same thing in a slightly different way and could be simplified.
Line 24-25 also needs re-writing for clarity and to improve sentence structure.
Abstract
Line 43 to end: I am not sure you can say this because how you are linking the preference for a rabbit’s features with potential health and welfare problems? Your study is not aiming to fix these problems, you have simply measured what people like. Are you assessing how much owners know about health problems in these short-faced rabbits? If so they you could have a stronger justification for saying that your research has a welfare outcome.
I think Line 45 is try- you have identified reasons why people buy pet rabbits, so perhaps you should pose a research extension (along health and welfare grounds) from this.
Similar issues with convoluted sentence style in the abstract as per the simple summary. Please try and write in a clearer and more easy-to-follow manner. I also suggest defining key technical terms (such as dolichocephalic).
Introduction
Line 49-50: Needs a supporting citation otherwise is your own presumption.
Line 50-51: Are you sure that “comparatively little is known about the breeding of pet rabbits…” I think you need context here. A lot is known about how to breed rabbits! I get what you are trying to say, but please think about phrasing.
Line 53-55: A very long sentence. You need some punctation after the reference for what brachycephalic is. I would split this up to make it easier to read.
Line 57: Probably better to say this has become the norm in domestic rabbit breeds. It is not a species.
Line 62: Needs a citation for these new mechanisms, also needs better punctuation to improve readability.
Line 67: needs a citation for the breed standards.
Line 68: Also needs a reference. There is a lot of assumption in the introduction. Please include evidence for the points you are making.
Line 71: manipulated through selective breeding.
Line 76: I would make more of a bridge between the introduction of the dental problems and long ears and your explanation in the following paragraph. As dental issues in rabbits are probably the biggest welfare concern, after inappropriate housing.
Line 80: Your description of the brachycephalic changes in a rabbit’s skull needs a citation. The same with the fact of the majority of domestic rabbits… are there figures for this? Who says it’s a majority?
Line 86: What’s a BC rabbit? One that’s prehistoric?! I know what you mean, but please define your abbreviations to the reader.
Line 89: Do you have any evidence for this growth in popularity?
Line 90-91: Again, needs a reference and a check of punctuation. “…popularity of, and selection for, …” would read more fluently.
Line 99: This is good. You explain precisely what is unexplored and therefore based on anecdote. Try and mirror this writing style in the other areas of your introduction where you need to suggest an effect but you don’t have any evidence.
Line 110-end of paragraph: Very long sentences, hard to follow. Can split these up and be more concise in the points you are making?
Also, mesocephalic is introduced with no definition.
Line 117: There is no bridge between the explanation of research on dogs and cats and the aims of your experiment. Can you use the dog and cat literature as more of a support for why you are now investigating rabbits.
Materials and methods
Line 128: “… authors’ own rabbits…”
Line 129: “… that were freely…”
Line 133: Can you explain what you mean by “balanced”?
Line 137: I appreciate the use of the reference to show that the method is based on precedent, but I do feel a brief explanation of what this is would be useful.
Line 153-155: Again, the scale should be explained rather than simply relying on the citation.
Line 156: You need to introduce brief definitions of the terms when they are first mentioned in the text too.
Line 161: Why only these three vet schools?
Line 176: Any precedent for the 0-3 and 7-10 being used for determining these categories of dislike and like?
Stats
Can you include some explanation of why statistical tests were chosen? I.e. I am assuming that data did not follow a normal distribution because medians are used and because KW and MW tests are included? Perhaps move Line 204 to the start of the paragraph?
Results
Line 210: How was expert defined?
I think Table 1 is more methods / sample population rather than a result.
Again, section 3.2, this is describing the sample population, so therefore should be presented in the methods before the results section.
Line 272-274: How were these variables controlled? I am struggling to understand this part of the results section. Where has ear length, fur colour etc. been explored earlier in your analysis the same as you have focussed on for head shape?
Are you able to present r2 values or AIC values to show the fit of your models? You could use the MuMIn package in R if your models have been run in R? It would be good to see how much variation is explained by some of these models.
Line 306: How was “fluffiness” defined and made measurable? However, I love the fact that you have included this as a term in your output ?
And further on in this paragraph, how can a rabbit be too fluffy? How was this standardised? Perhaps more explanation of the categories in the questionnaire, as supplementary information, is needed?
Figure 2 and Figure 3: Please label your axes and provide units. I really do not understand what Figure 3 is showing me. Figure 2 seems logical.
You tabulate lots of multiple P values from your model output. Would it not be worth applying a correction factor to check for Type I error by comparing your P values to a new significance level? E.g. Benjamini-Hochberg or similar?
Discussion
Please start the discussion with an overall explanation of your results. This would be more relevant to the reader than a description of domestic rabbits have flatter faces that you have mentioned several times previously early in the manuscript.
Can you link back to tables and figures in your results that show the points you are making in your discussion? I.e. we show that this type of rabbit is really popular with students (see Table X). You have a lot of results, which is good, so it would help direct the reader in your discussion.
Line 352: I appreciate this is in an interesting study and you have had a large number of responses but I don’t think you need to state that it is ground-breaking. Simply say that the study attracted an impressive level of response of XXXX across XXX countries or similar.
Line 388: “these data…”
Line 402: Please give a reference (baby-schema).
Line 414: The RCVS considers rabbits an exotic species. Is this the case all over the world? Please provide a reference.
Also for Line 414, ALL animals have specific husbandry and welfare requirements. Not just exotic ones. Consider your turn of phrase.
Line 415-416: Maybe “… without how to care for basic and breed-specific needs”?
Line 416: Did you measure reasons for purchasing a rabbit? If so can you support these statements with your data?
Line 425: provide a reference for this higher risk.
Line 438: Do Netherland dwarf rabbits suffer from many health issues? Could you expand on your evaluation here using your own data and extrapolating to an extension of this project?
Conclusion
Line 488: If you explain BC the first time it is mentioned, you don’t need to repeat it here and you can just use the abbreviation.
Line 490: erect-eared
I understand that there is useful information here to inform people’s choice. But you haven’t measured how much people know about health conditions and face shape. So I would re-write some of your discussion to say that “now we know why people choose what they choose, we can educate potential owners on the problems of such an animal and suggest an alternative” Or something like this.
Author Response
We would like to thank the reviewer for their time and have amended the manuscript accordingly where possible. However, we feel like many of these comments are issues of personal preference or opinion from the reviewer (particularly regarding writing style), which do not match with the other two reviewers’ feedback, and so we have not made changes for every comment.
Detail review below:
Simple summary
The simple summary needs to be reworded because it contains numerous descriptions of the rabbits that are not really quantified or explained. what’s the difference, for example, between a flat-faced rabbit and a midly-flat-faced rabbit? How has this been measured?
This has now been changed to: “mildly-flat-faced rabbits (on a scale from extremely-flat-faced to extremely-long-faced)”
Line 17 and 18 appears to say the same thing in a slightly different way and could be simplified.
This has been changed now.
Line 24-25 also needs re-writing for clarity and to improve sentence structure.
Some additional wording has been added here.
Abstract
Line 43 to end: I am not sure you can say this because how you are linking the preference for a rabbit’s features with potential health and welfare problems? Your study is not aiming to fix these problems, you have simply measured what people like. Are you assessing how much owners know about health problems in these short-faced rabbits? If so they you could have a stronger justification for saying that your research has a welfare outcome.
We have altered this section now.
I think Line 45 is try- you have identified reasons why people buy pet rabbits, so perhaps you should pose a research extension (along health and welfare grounds) from this.
Similar issues with convoluted sentence style in the abstract as per the simple summary. Please try and write in a clearer and more easy-to-follow manner. I also suggest defining key technical terms (such as dolichocephalic).
Dolichocephalic has now been defined.
Introduction
Line 49-50: Needs a supporting citation otherwise is your own presumption.
The authors feel that this is a generic statement, and therefore does not need a citation. Anyone can find this out with a quick search on PubMed for example, where there is a single paper on breeding of pet rabbits compared to 754 hits for breeding dogs or 318 for breeding cats. This has now been made clear in the text (see below comment response).
Line 50-51: Are you sure that “comparatively little is known about the breeding of pet rabbits…” I think you need context here. A lot is known about how to breed rabbits! I get what you are trying to say, but please think about phrasing.
Please see above response. The paper we cited at the end of this sentence was one of the first to examine pet rabbit breeding. This has now been changed to this: “When considering the health and welfare implications of companion animal breeding, most research and public discourse concerns the impacts on cats and dogs. For example, only a single study has been published about the breeding of pet rabbits[1], despite being the third most popular companion animal (excluding fish) in the United Kingdom (UK) [4].”
Line 53-55: A very long sentence. You need some punctation after the reference for what brachycephalic is. I would split this up to make it easier to read.
Extra punctuation has now been added here, thank you.
Line 57: Probably better to say this has become the norm in domestic rabbit breeds. It is not a species.
Changed.
Line 62: Needs a citation for these new mechanisms, also needs better punctuation to improve readability.
Punctuation has been added. This is our own theory that we have suggested here, it is not something that requires citing. The wording has been changed slightly to make this clearer.
Line 67: needs a citation for the breed standards.
The BRC standards have now been cited here.
Line 68: Also needs a reference. There is a lot of assumption in the introduction. Please include evidence for the points you are making.
There isn’t any evidence, as per our point earlier. I would cite something if I could, but there is very little published study on pet rabbits. I have added ‘authors personal observation’ now to make this clear.
Line 71: manipulated through selective breeding.
Thank you, this has now been added.
Line 76: I would make more of a bridge between the introduction of the dental problems and long ears and your explanation in the following paragraph. As dental issues in rabbits are probably the biggest welfare concern, after inappropriate housing.
We feel that the paragraphs flow well as they are so have not made any changes. If we were to focus more on dental health issues, even though they are a common health concern, this could detract from the main focus of the paper which is perceptions/preferences of morphological characteristics. Additionally, dental problems are a multifactorial issue, with diet and housing being substantial contributors, expansion into these areas is beyond the intent of this paper and is a research question in its own right. Apologies if we have misunderstood the reviewer’s comment.
Line 80: Your description of the brachycephalic changes in a rabbit’s skull needs a citation. The same with the fact of the majority of domestic rabbits… are there figures for this? Who says it’s a majority?
The changes to animals’ skull shape that occur in brachycephaly does not require a reference in our opinion; it is not new knowledge it is simply what brachycephaly is. With regard to your comment about the ‘majority’ statement, it is cited at the end of the sentence.
Line 86: What’s a BC rabbit? One that’s prehistoric?! I know what you mean, but please define your abbreviations to the reader.
It’s defined in line 54 J
Line 89: Do you have any evidence for this growth in popularity?
There is quantified evidence in dogs as they are registered via kennel clubs, but there is no quantified evidence for other animals. References to canine evidence has now been added.
Line 90-91: Again, needs a reference and a check of punctuation. “…popularity of, and selection for, …” would read more fluently.
Thank you, the punctuation has now been added. The reference, which was at the end of the following sentence, has now been moved here.
Line 99: This is good. You explain precisely what is unexplored and therefore based on anecdote. Try and mirror this writing style in the other areas of your introduction where you need to suggest an effect but you don’t have any evidence.
Line 110-end of paragraph: Very long sentences, hard to follow. Can split these up and be more concise in the points you are making?
This has now been split.
Also, mesocephalic is introduced with no definition.
This has now been added in.
Line 117: There is no bridge between the explanation of research on dogs and cats and the aims of your experiment. Can you use the dog and cat literature as more of a support for why you are now investigating rabbits.
I have added an additional bridging sentence here now.
Materials and methods
Line 128: “… authors’ own rabbits…”
Corrected.
Line 129: “… that were freely…”
Corrected.
Line 133: Can you explain what you mean by “balanced”?
The same number of each type in each group – which was of course a guess as we didn’t have the cephalic ratings at that point, but we did try to balance the phenotypic features across skull shapes. More description has now been added here.
Line 137: I appreciate the use of the reference to show that the method is based on precedent, but I do feel a brief explanation of what this is would be useful.
Brief description of the studies purpose has now been added.
Line 153-155: Again, the scale should be explained rather than simply relying on the citation.
The scale is explained in parenthesis: “a scale that ranged from 1 – 7 (1= extreme brachycephalism (BC), 2 = moderate BC, 3 = mild BC, 4 = mesocephalic (MC), 5 = mild dolichocephalism (DC), 6 = moderate DC and 7 = extreme DC)”
Line 156: You need to introduce brief definitions of the terms when they are first mentioned in the text too.
Brachycephalic is defined in line 54, but I have now added definitions of mesocephalic and dolichocephalic into line 125.
Line 161: Why only these three vet schools?
These are the only ones the authors had contacts at who were able to get the survey shared amongst the staff and students.
Line 176: Any precedent for the 0-3 and 7-10 being used for determining these categories of dislike and like?
No, sadly not. These just represent the extreme ends of the scale.
Stats
Can you include some explanation of why statistical tests were chosen? I.e. I am assuming that data did not follow a normal distribution because medians are used and because KW and MW tests are included?
Medians were used to the cephalic rating’s so that the images could be categorised into a cephalic group, which was more of a functional factor than a statistical one. The preference score data however was slightly right skewed towards higher preference scores, but some could argue it still approximated normal. Some statisticians argue Likert scale data isn’t truly continuous data, so should be analysed with non-parametric statistics – but other statistician’s argue parametric statistics are fine for Likert data, so even full statisticians can’t agree. There’s no real downside to using non-parametric tests though unless you have a very small sample size, as the tests are just slightly more conservative, which for this study really didn’t matter. There’s a good discussion of this with some references here: https://www.theanalysisfactor.com/can-likert-scale-data-ever-be-continuous/
Perhaps move Line 204 to the start of the paragraph?
Done.
Results
Line 210: How was expert defined?
It wasn’t. The words (as can be seen in Table A1) were ‘Would you consider yourself a rabbit specialist”, with the answers yes/no/not sure. I have changed the word ‘expert’ to ‘specialist’ in the text now.
I think Table 1 is more methods / sample population rather than a result.
There’s no solid convention for where to report the study population (methods or results). Here, we reported how we distributed the survey in the methods, and we consider part of the results to be how many and what type of responses we received. I haven’t changed this
Again, section 3.2, this is describing the sample population, so therefore should be presented in the methods before the results section.
See previous comment.
Line 272-274: How were these variables controlled? I am struggling to understand this part of the results section. Where has ear length, fur colour etc. been explored earlier in your analysis the same as you have focussed on for head shape?
Are you able to present r2 values or AIC values to show the fit of your models? You could use the MuMIn package in R if your models have been run in R? It would be good to see how much variation is explained by some of these models.
The models were run in MLwiN. MLwiN uses quasilikelihood estimation to fit these models, so AIC statistics and R-squared values are not appropriate and are not produced unfortunately.
Line 306: How was “fluffiness” defined and made measurable? However, I love the fact that you have included this as a term in your output ?
Sadly, it really isn’t measurable or defined, but is a lay-term that was used in the survey, so it reflects people’s subjective judgement of the rabbits fur as ‘fluffy’
And further on in this paragraph, how can a rabbit be too fluffy? How was this standardised? Perhaps more explanation of the categories in the questionnaire, as supplementary information, is needed?
Again, this was just people answering to say that for them it was too fluffy. The full list of answers they responded to are in Appendix B, Table B2 as referenced in section 2.2.
Figure 2 and Figure 3: Please label your axes and provide units. I really do not understand what Figure 3 is showing me. Figure 2 seems logical.
I honestly don’t know how to make figure 3 any clearer I’m afraid. I’ve added an axis label to it now, but I don’t see how that will help as it says the same thing the legend did. For figure 2 we do not feel that it would be appropriate to have a Y axis as these percentages aren’t meant to be summable (as they are from different categories of image), so it would convey no additional meaning. The two graphs are on the same scale though. This graph isn’t perfect but seemed to be the best way to visually represent the differing data.
You tabulate lots of multiple P values from your model output. Would it not be worth applying a correction factor to check for Type I error by comparing your P values to a new significance level? E.g. Benjamini-Hochberg or similar?
I would usually apply a correction for multiple testing, however in this case none of the significant p-values were anywhere near the threshold of p<0.05, so correcting is not likely to change anything. It would be simpler and have the same outcome to simply set a threshold at p<0.001, which I have done now and written into the end of the methods.
Discussion
Please start the discussion with an overall explanation of your results. This would be more relevant to the reader than a description of domestic rabbits have flatter faces that you have mentioned several times previously early in the manuscript.
This is only a single sentence which I feel places a beginning to this paragraph, which is followed by a middle and end sentence nicely introducing the key message. It doesn’t impact the scientific soundness of the study and to be honest, I would really rather not change this as it took a long time for me to find an appropriate way to begin this discussion, and I quite like it, I hope you understand.
Can you link back to tables and figures in your results that show the points you are making in your discussion? I.e. we show that this type of rabbit is really popular with students (see Table X). You have a lot of results, which is good, so it would help direct the reader in your discussion.
It's not normal procedure to reference back to tables or figures in discussions, so I don’t feel like this is necessary.
Line 352: I appreciate this is in an interesting study and you have had a large number of responses but I don’t think you need to state that it is ground-breaking. Simply say that the study attracted an impressive level of response of XXXX across XXX countries or similar.
We have changed this to ‘an exceptional level of engagement’.
Line 388: “these data…”
Thank you, corrected now.
Line 402: Please give a reference (baby-schema).
This was already referenced in the introduction, but I have re-referenced it again here now.
Line 414: The RCVS considers rabbits an exotic species. Is this the case all over the world? Please provide a reference.
Amended and citation added.
Also for Line 414, ALL animals have specific husbandry and welfare requirements. Not just exotic ones. Consider your turn of phrase.
This line has been altered to clarify.
Line 415-416: Maybe “… without how to care for basic and breed-specific needs”?
Thank you, this has now been altered.
Line 416: Did you measure reasons for purchasing a rabbit? If so can you support these statements with your data?
No, we didn’t ask about purchasing behaviour.
Line 425: provide a reference for this higher risk.
The references have already been described earlier in the text, this is simply a reminder sentence for what has already been discussed. I have now added “(as discussed earlier)” to this sentence.
Line 438: Do Netherland dwarf rabbits suffer from many health issues? Could you expand on your evaluation here using your own data and extrapolating to an extension of this project?
Anecdotally, yes, certainly if you speak to rabbit vets. However, data on rabbits in veterinary practice at the breed level is scarce, and breed isn’t often even recorded beyond ‘lop’ or ‘dwarf’.
Conclusion
Line 488: If you explain BC the first time it is mentioned, you don’t need to repeat it here and you can just use the abbreviation.
Corrected, thank you.
Line 490: erect-eared
Corrected.
I understand that there is useful information here to inform people’s choice. But you haven’t measured how much people know about health conditions and face shape. So I would re-write some of your discussion to say that “now we know why people choose what they choose, we can educate potential owners on the problems of such an animal and suggest an alternative” Or something like this.
We feel like this point is covered in the Discussion lines 475 to 486. However, we have altered the final line of the conclusion to this affect.
Reviewer 2 Report
Summary
The aim of this paper it is evaluating several phenotypic parameters from domestic rabbits and its correlation with public rating. Their main contributions are showing public preferences for mildly brachycephalic rabbits, stable across continents and with a large sample of individuals.
Broad comments
The study addresses an important subject which are pathological phenotypes in rabbits which is scarcely found in the literature. The conclusions are supported by the data and as suggested by authors would be a useful tool for improving education and sensitization for breeders, owners and authorities about pathological phenotypes in rabbits.
I found the article well organized and clear in the presented manner. However, I suggest for clarity start the introduction (P1 L49) with the general issue, "impact of animal breeding" and not directly with limitations of previous research (limited to dogs and cats) which can be addressed later in the same paragraph. In the discussion, the authors talk about decreased “an average of one point” in Europe and Oceania, however (L360) this comment can’t be clearly related to the results showed and it creates some misunderstanding reading table S4. To clarify I suggest comment only results showed in the paper and additional documents, which show one point less median in Europe, and in higher and lower interquartile limits, and one point less in lower interquartile limits for Oceania.Specific comments
L148 appendix B, Table A1?L160 has been the questionnaire validated? please complete this information within the text.
L362 for clarity: It would be easier to understand “differences between continents"
or "between-continents differences"
L363 "from Europe and Australasia" instead of "Europe and Australasia)"
Please, be consistent with nomenclature, choose Oceania or Australasia (L363).
L373 I don't think cultural differences and media are exclusive or alternatives, media coverage and campaigning is part of culture continuous shaping and therefore in society perception. I suggest relating both hypotheses.
Author Response
Reviewer 2
The aim of this paper it is evaluating several phenotypic parameters from domestic rabbits and its correlation with public rating. Their main contributions are showing public preferences for mildly brachycephalic rabbits, stable across continents and with a large sample of individuals.
Broad comments
The study addresses an important subject which are pathological phenotypes in rabbits which is scarcely found in the literature. The conclusions are supported by the data and as suggested by authors would be a useful tool for improving education and sensitization for breeders, owners and authorities about pathological phenotypes in rabbits.
I found the article well organized and clear in the presented manner. However, I suggest for clarity start the introduction (P1 L49) with the general issue, "impact of animal breeding" and not directly with limitations of previous research (limited to dogs and cats) which can be addressed later in the same paragraph.
Thank you so much for your kind and constructive comments! The opening paragraph has now been restructured.
In the discussion, the authors talk about decreased “an average of one point” in Europe and Oceania, however (L360) this comment can’t be clearly related to the results showed and it creates some misunderstanding reading table S4. To clarify I suggest comment only results showed in the paper and additional documents, which show one point less median in Europe, and in higher and lower interquartile limits, and one point less in lower interquartile limits for Oceania.
Well spotted, this was written in error and has now been corrected.
Specific comments
L148 appendix B, Table A1?
Thank, you, this has now been corrected.
L160 has been the questionnaire validated? please complete this information within the text.
No, as we didn’t have the resources to do it. This has been made clear here now. Creating this survey was the only option for ranking the rabbits according to cephalic degree as there is no cephalic index for rabbits.
L362 for clarity: It would be easier to understand “differences between continents"
or "between-continents differences"
Changed
L363 "from Europe and Australasia" instead of "Europe and Australasia)"
Please, be consistent with nomenclature, choose Oceania or Australasia (L363).
‘Australasia’ was how the manuscript we referenced here referred to it, so we used their nomenclature, I have changed this to Oceania now though in response to your request.
L373 I don't think cultural differences and media are exclusive or alternatives, media coverage and campaigning is part of culture continuous shaping and therefore in society perception. I suggest relating both hypotheses.
Good point, this has now been corrected to: “could be due to cultural differences in perceptions, which may be linked to media coverage and campaigning for awareness of issues associated with brachycephaly”
Reviewer 3 Report
The presented manuscript investigates how head morphology of rabbits affects the preference of humans, via a questionnaire study. Globally, participants in the study preferred mildly brachycephalic rabbits the most, and moderately-dolichocephalic rabbits the least. This quite likely could be a major driver for the shortening of the head typically seen in domestic rabbits. The insights of the presented study are relevant because brachycephaly is associated with considerable health problems and knowledge about the preference of humans could inform educational programs, informing the general public and breeders about the implications of their preferences. Hopefully, being informed about the implications of their choices with regards to the welfare of their pets, pet owners would start to act against their preference with regards to the optics of the animals. Overall the presented study is timely and of interest for a broad readership in the area of pet animal welfare.
Overall, I find the manuscript extremely well written and would like to congratulate the authors to this. One minor concern, throughout the manuscript, the authors refer to ‘skull morphology’, however they use photographs of the life rabbit. I am wondering to what degree the head shape is really correlated with skull shape and I guess other factors like muscular development strongly affect the head shape. I would like to see a paragraph in the introduction either validating the statement that skull morphology and head shape are relatively synonymous or alternatively use head shape or a similar term throughout the manuscript, instead of skull morphology.
Another comment relates to the results section. I think the wording in the results section could be improved to clearly link the research question with the parameters recorded. At times, the authors phrase their results more in statistical terms, and do not clearly guide the reader through the patterns in the recorded answers. What do answers mean, what do statistically significant results mean?
Ln. 51-52: Why is the ‘excluding fish’ relevant? Does this mean fish would be the third most popular? Does the ranking then refer only to mammalian species, or mammals, birds and reptiles? Please clarify.
Lns. 147 and 169: Has personal information been collected in the survey, in which case it is not anonymous, but confidential. I think it needs to be clarified what ‘informed consent’ means in this case, did you ask for participants names or signature, or did they just need to tick a box to confirm consent? If you collected personal information (names and signatures), please comment on what has been done to ensure data protection (how was personal information stored, who has access to it? How long was it stored)?
Lns. 185- 206: The statistical analysis seems a bit exhaustive to me and the method to reduce factors applying prior pair-wise comparisons (Mann-Whitney or Kruskal-Wallis tests) seems very unusual. I am more familiar with stepwise selection procedures or AICc model selection, to reduce number of factors (which all are not without criticism). Another opportunity could be to present the full model, or full and reduced models.
I am not an expert for statistical analysis of questionnaires, but I am wondering if the chosen methods are really appropriate. The pair-wise comparisons seem to result in a lot of significant patterns and I am wondering whether all of these patterns are really meaningful and how much they are just a result from the large sample size?
Lns. 209-214: Have all veterinary professions been English (British)? Has this information been collected?
Lns. 234-241: Move reference to table 3 either at the beginning or end of the paragraph as the table summarizes the entire paragraph, rather than only one statement within it.
Table 2: As a suggestion, would it be possible to add the actual pictures in the table? I personally think it would be quite beneficial for the reader.
Ln. 266 and 268: This should probably be table 4?
Ln. 279: n should be N
Figure 2: has the description of the Y-axis missing. I personally do not think the figure heading (liked because, disliked because) is needed.
Figure 3: not sure if this is correctly displayed in my PDF. I do not see any bars, but only percentages. If the authors intend to only present the percentages, this could be better done in a table.
Ln. 352: I agree that the authors achieved a very impressive participation rate in their questionnaire study, however the terminology ‘ground-breaking level of engagement’ seems a bit exhaustive and too conversational for my taste. Consider rephrasing, e.g. impressive, large, larger than usually presented in comparative studies (present references).
Author Response
Thank you for your very kind comments!
Re: Skull morphology, the skull shape is the only mechanism by which the muzzle length varies. Face shape can only be affected by underlying skull shape and cannot occur solely through muscular differences, which even if present, would in turn change the skull morphology. Lop ears (a soft tissue) for example still correlate with changes in skull morphology. We haven’t made any changes in response to this as we feel this isn’t necessary.
Another comment relates to the results section. I think the wording in the results section could be improved to clearly link the research question with the parameters recorded. At times, the authors phrase their results more in statistical terms, and do not clearly guide the reader through the patterns in the recorded answers. What do answers mean, what do statistically significant results mean?
I have read through the results section carefully and tried to find areas to improve, which I have altered where possible, but struggled to identify which areas aren’t clear. Co-authors have also checked this point, but again, we don’t know what further to change here.
Ln. 51-52: Why is the ‘excluding fish’ relevant? Does this mean fish would be the third most popular? Does the ranking then refer only to mammalian species, or mammals, birds and reptiles? Please clarify.
It refers to companion animals, which aren’t typically considered to be fish. Additionally, the numbers of fish are not quantified, so we cannot include them in any ranking. A single person can own hundreds of fish, making them likely to be the most commonly owned pet. But their numbers are unsubstantiated as companion animals. We could remove the reference to fish altogether, but I’d rather it was clear that we have not included them.
Lns. 147 and 169: Has personal information been collected in the survey, in which case it is not anonymous, but confidential. I think it needs to be clarified what ‘informed consent’ means in this case, did you ask for participants names or signature, or did they just need to tick a box to confirm consent? If you collected personal information (names and signatures), please comment on what has been done to ensure data protection (how was personal information stored, who has access to it? How long was it stored)?
Well spotted, we have now changed to include the relevant information. Thank you! We collected no names, or signatures, they just ticked a box to consent to the survey in terms of providing their answers and the fact that as it was anonymous we would not be able to delete any answers upon request. However, at the end of the survey there was an option for them to tick a box to say if they wanted to leave an email address to be informed about the study results once they are published. At that point additional consent was gained (by ticking a box again) to agree to use storing their email for that purpose only, after which it will be deleted. The email was the only ‘personal data’ collected, and only for those who wanted to provide it. So they survey was anonymous, but they could confidentially provide an email address at the end. This has now been added in section 2.
Lns. 185- 206: The statistical analysis seems a bit exhaustive to me and the method to reduce factors applying prior pair-wise comparisons (Mann-Whitney or Kruskal-Wallis tests) seems very unusual. I am more familiar with stepwise selection procedures or AICc model selection, to reduce number of factors (which all are not without criticism). Another opportunity could be to present the full model, or full and reduced models.
This is a standard method employed for this type of analysis. It’s actually considered better practice to pre-filter rather than relying purely on stepwise methods which have their own issues as well as you allude to. This method of analysis matches that in the original paper on cats (Farnworth et al 2018), which we based this study on, which means the results are most directly comparable.
I am not an expert for statistical analysis of questionnaires, but I am wondering if the chosen methods are really appropriate. The pair-wise comparisons seem to result in a lot of significant patterns and I am wondering whether all of these patterns are really meaningful and how much they are just a result from the large sample size?
The pair-wise comparisons are in the supplementary material only and we don’t draw any conclusions from these, we just present the results for transparency. The p-value will of course react to the sample size, however, the Z value and term coefficient, can be used as a guide for the strength of association, which should not be impacted by the sample size, and is presented alongside the p-value. P-values never tell anyone what is meaningful. I’ve had to correct people during review before when they’ve tried to make a big thing out of a significant correlation, where the rho was 0.09, but p was <0.05 simply because of an enormous sample. However, a rho of 0.09 is not meaningfully different from zero. In this case, the coefficients do vary in strength, and the smallest ones are not ‘significant’ so I am comfortable with these results.
Lns. 209-214: Have all veterinary professions been English (British)? Has this information been collected?
This information was not collected as it was not considered to be relevant.
Lns. 234-241: Move reference to table 3 either at the beginning or end of the paragraph as the table summarizes the entire paragraph, rather than only one statement within it.
This had now been moved to the beginning.
Table 2: As a suggestion, would it be possible to add the actual pictures in the table? I personally think it would be quite beneficial for the reader.
I agree this would be lovely, but when I tried to do it, the table became far too long (pages long) unless the images were so small that they could barely be seen, which defies the point. I do agree this would be better though!
Ln. 266 and 268: This should probably be table 4?
Well spotted! All table numbers have now been updated, thank you.
Ln. 279: n should be N
Corrected
Figure 2: has the description of the Y-axis missing. I personally do not think the figure heading (liked because, disliked because) is needed.
We haven’t changed this figure. These percentages aren’t summable (as they are from different categories of image), so a Y axis wouldn’t be appropriate as it conveys no real meaning. The two graphs are on the same scale though. This graph isn’t perfect but seemed to be the best way to represent the differing data. The figure heading comment is a valid personal opinion, but we feel like they are helpful for readers to differentiate the data more easily.
Figure 3: not sure if this is correctly displayed in my PDF. I do not see any bars, but only percentages. If the authors intend to only present the percentages, this could be better done in a table.
I don’t know what happened here as in the PDF I uploaded the bars are there. I can only assume it was an uploading error of some sorts, which I hope won’t happen again.
Ln. 352: I agree that the authors achieved a very impressive participation rate in their questionnaire study, however the terminology ‘ground-breaking level of engagement’ seems a bit exhaustive and too conversational for my taste. Consider rephrasing, e.g. impressive, large, larger than usually presented in comparative studies (present references).
We have changed this to ‘an exceptional level of engagement’ now.